# Influence of Biomass Inorganics on the Functionality of H⁺ZSM-5 Catalyst during In-Situ Catalytic Fast Pyrolysis

**Ravishankar Mahadevan, Sushil Adhikari \*, Rajdeep Shakya and Oladiran Fasina**

Department of Biosystems Engineering, Auburn University, Auburn, AL 36849, USA;
ravishankar.mahadevan@intel.com (R.M.); rzs0035@tigermail.auburn.edu (R.S.); fasinoo@auburn.edu (O.F.)
\* Correspondence: sushil.adhikari@auburn.edu; Tel.: +1-334-844-3543; Fax: +1-334-844-3530

**Abstract:** In this study, the contamination of H⁺ZSM-5 catalyst by calcium, potassium and sodium was investigated by deactivating the catalyst with various concentrations of these inorganics, and the subsequent changes in the properties of the catalyst are reported. Specific surface area analysis of the catalysts revealed a progressive reduction with increasing concentrations of the inorganics, which could be attributed to pore blocking and diffusion resistance. Chemisorption studies ($NH_3$-TPD) showed that the Bronsted acid sites on the catalyst had reacted with potassium and sodium, resulting in a clear loss of active sites, whereas the presence of calcium did not appear to cause extensive chemical deactivation. Pyrolysis experiments revealed the progressive loss in catalytic activity, evident due the shift in selectivity from producing only aromatic hydrocarbons (benzene, toluene, xylene, naphthalenes and others) with the fresh catalyst to oxygenated compounds such as phenols, guaiacols, furans and ketones with increasing contamination by the inorganics. The carbon yield of aromatic hydrocarbons decreased from 22.3% with the fresh catalyst to 1.4% and 2.1% when deactivated by potassium and sodium at 2 wt %, respectively. However, calcium appears to only cause physical deactivation.

**Keywords:** biomass ash; catalytic fast pyrolysis (CFP); hydrocarbons; in-situ; deactivation; bio-oil

## 1. Introduction

Catalytic fast pyrolysis (CFP) has been investigated in recent years as a thermochemical conversion method for producing partially deoxygenated liquid fuel intermediates from biomass. Compared to the quality of oil produced from non-catalytic pyrolysis, the use of heterogeneous catalysts instead of inert heat carriers during in-situ CFP results in the removal of oxygen and the production of a liquid product (bio-oil) containing a higher heating value and reduced oxygen content [1,2]. The catalytic deoxygenation of a wide range of compounds produced from pyrolysis such as ketones, aldehydes, alcohols, carboxylic acids and phenolics reduces the extent of subsequent upgrading required before the bio-oil could be used as a drop-in fuel [3,4]. The improved thermal stability and lower oxygen content of bio-oil produced from CFP decreases the burden on the economically inefficient hydrotreating step, which utilizes expensive metal catalysts, high temperature and high pressure of hydrogen [5–8]. Solid acid catalysts such as H⁺ZSM-5, Y-zeolite, β-zeolite are among the most commonly used materials, which transform the pyrolysis vapor by rejecting oxygen through dehydration ($-H_2O$), decarboxylation ($-CO$) and decarbonylation ($-CO_2$) reactions, leading to a product composed of aromatic hydrocarbons and olefins.

The zeolite family of catalysts are particularly interesting for CFP due to the presence of a large number of Bronsted (O.H.) and Lewis (≡Al) acid sites, which are present simultaneously and catalyze a number of cracking reactions with a high selectivity for producing olefins and aromatic hydrocarbons [9,10]. However, the presence of a large number of acid sites leads to the formation of coke on the surface and in the pores, resulting in a gradual loss in catalytic activity and carbon yield of hydrocarbons. Pyrolysis reactor systems typically counter this problem by employing a regeneration step wherein the coked catalyst

is thermally oxidized to remove the carbon deposits in an effort to restore the original activity of the catalyst and provide process heat for the pyrolysis zone using the exothermic nature of the oxidation reaction. Among these catalysts, $H^+$ZSM-5 zeolite has been the most widely studied and considered unique due to its shape selectivity that suppresses the coke formation, while also maximizing the conversion to aromatic hydrocarbons [11].

The catalytic conversion of biomass via CFP also faces several challenges due to the inherent complexity of the feedstock, which can vary in its composition (i.e., cellulose, hemicellulose and lignin content), leading to large variations in the yield and distribution of the products, and making it difficult to consistently produce a product of uniform quality. Many studies have been performed to investigate and understand the effect of such varying composition of the biomass constituents on non-catalytic pyrolysis [12–20]. The effect of inorganic constituents of the biomass ash, which contains alkali and alkaline earth metals (AAEMs) such as sodium, potassium, calcium and magnesium has also been investigated for non-catalytic pyrolysis [15,21–30]. The presence of these inorganics in the biomass has been observed to be detrimental due to its influence on the pyrolysis product distribution and yield, promoting the formation of lower molecular weight cellulose and lignin derivatives while increasing the yield of thermally-derived char and non-condensable gases, as reported in a recent study from our group [31].

Besides causing changes to the pyrolysis mechanism, these metals are mostly retained in the char product after pyrolysis and are usually circulated along with the catalyst to the regeneration reactor. The high temperatures (> 650 °C) employed in this zone are sufficient to vaporize the inorganics and these metals have been observed to accumulate on the catalyst by several studies in the literature [11,32–35]. The AAEMs might cause chemical poisoning by reducing the number of acid sites or physical poisoning by blocking the pore mouth, increasing the diffusion resistance and resulting in less accessibility to the active sites of the catalyst. This type of deactivation with AAEMs cannot be reversed without the use of inorganic acids to remove the contaminants and results in permanent deactivation of the catalyst, affecting the economic feasibility of the process. Mullen et al. [34] studied the accumulation of various inorganics on $H^+$ZSM-5 with a $SiO_2/Al_2O_3$ ratio of 30 during the in-situ CFP of switchgrass and reported the linear accumulation of the total amounts of Ca, Cu, Fe, K, Mg, Na and P on the $H^+$ZSM-5 surface with increasing exposure of the catalyst to biomass. The accumulation of inorganics on the catalyst was correlated to the drop in catalyst activity for producing deoxygenated product, resulting in a decreased yield of aromatic hydrocarbons and the lower carbon to oxygen ratio. Yildiz et al. [32,36] reported the accumulation up to 3 wt % of ash from pinewood biomass on the catalyst ($H^+$ZSM-5) and also observed changes in the distribution and composition of the products from pyrolysis. The conversion of sugars, acids and phenols were suppressed due the presence of higher concentrations of accumulated ash. Paasikallio et al. [35] investigated the in-situ CFP of pine sawdust and observed a small increase in the oxygen content of the bio-oil (22.4–23.7 wt %) over the course of a four day pyrolysis run, while the catalyst retained most of its original activity during the experimental period. Stefanidis et al. [33] studied hydrothermal deactivation and metal contamination during in-situ CFP of commercial beech wood biomass using $H^+$ZSM-5 catalyst and reported that the individual rates of accumulation of metals were different, with some metals like potassium accumulating more readily than others. They also reported that hydrothermal deactivation and metal contamination led to a linear loss in catalyst activity during pyrolysis, resulting in the production of bio-oil with higher oxygen contents (from 20 to 35 wt %). While the results of these studies suggest the overall correlation between the loss in catalyst activity and the accumulation of inorganics on the catalyst, it is not clear since other factors such as catalyst attrition, loss in surface area during the course of the experiment and decreasing catalyst to biomass ratios were not controlled. Another important consideration for performing this study is the wide variation in the composition of inorganic species in different types of biomass available in various parts of the world. Investigating the individual influence of the inorganic species on the catalyst would help to develop a fundamental understanding

of their effect on the product distribution and quality from CFP and would eliminate some of the uncertainties associated with utilizing biomass.

In this study, we investigated the effect of individual biomass inorganics on the functionality of $H^+$ZSM-5 catalyst during in-situ CFP. It was performed on the hypothesis that different inorganic minerals (Ca, K and Na) would have varied influence on the functionality of the catalyst depending on the level of contamination and the type of mineral. ZSM-5 catalysts were impregnated separately with different concentrations (0.5, 1, 2 and 5 wt %) of Ca, K and Na and characterized to study the impact of doping the metals on the properties of the catalyst. The catalysts deactivated by metal impregnation were subsequently used in in-situ CFP experiments in a micro reactor (pyroprobe) coupled with a GC–MS to understand the effect of metal contamination on the product distribution from CFP.

## 2. Results and Discussion

### 2.1. Biomass Characterization

The biomass used in this study (southern pine) was characterized to measure ash, moisture, higher heating value (HHV) and elemental composition, and summarized in Table 1. Inherent ash content of the biomass (0.63 wt %) was not washed or removed and ICP analysis revealed that it was composed of 0.2% calcium, 0.15% magnesium, 0.11% potassium and 0.06% sodium [31]. Cellulose, hemicellulose and lignin contents of the biomass were determined and shown in Table 2.

**Table 1.** Proximate and ultimate analyses of biomass used in this study.

| Proximate Analysis, As Received | Analytical Standard | Result—Pine |
|---|---|---|
| Ash content, wt % | ASTM E1755 | $0.63 \pm 0.07$ |
| Volatile Matter, wt % | ASTM E872 | $77.26 \pm 0.32$ |
| Moisture content, wt % | ASTM E871 | $6.44 \pm 0.53$ |
| Fixed carbon, wt % | By balance | $15.67 \pm 0.48$ |
| Heating Value, M.J./kg | ASTM E870 | $18.31 \pm 0.21$ |
| **Ultimate Analysis** | **Analytical Instrument** | **Pine** |
| C, wt % | | $45.69 \pm 0.29$ |
| H, wt % | | $6.63 \pm 0.08$ |
| N, wt % | Perkin-Elmer, model CHNS/O 2400 | $0.30 \pm 0.09$ |
| S, wt % | | $0.12 \pm 0.01$ |
| O, wt % | By difference | $46.97 \pm 0.13$ |

The number after $\pm$ denotes standard deviation. Ultimate analysis is in dry, ash-free basis.

**Table 2.** Results of component analysis, wt %, dry basis.

| Sample | Cellulose % | Hemicellulose % | | | | | Lignin % | | Extractives % |
|---|---|---|---|---|---|---|---|---|---|
| | | Xylan | Galactan | Arabinan | Mannan | Total | AIL | ASL | |
| Pine | $40.93 \pm 0.82$ | $7.38 \pm 0.13$ | $3.08 \pm 0.05$ | $1.52 \pm 0.014$ | $10.97 \pm 0.09$ | $22.96 \pm 0.29$ | $28.82 \pm 0.42$ | $1.83 \pm 0.07$ | $3.08 \pm 0.04$ |

Number followed by $\pm$ sign represents standard deviation.

### 2.2. Effect of Biomass Inorganics on the Properties of HZSM-5

BET (Brunauer–Emmett–Teller) specific surface area of the $H^+$ZSM-5 catalysts with different loadings of inorganic metals is shown in Figure 1, and it can be seen that the specific surface area decreases linearly with an increase in the concentration of inorganics impregnated in the catalyst. This finding is important even though the reduction in the specific surface area might not translate directly to the activity of the catalyst. However, reduction in the surface area decreases the accessibility of biomass molecules to the active sites in the catalyst during pyrolysis, and also increases diffusion resistance due to blocking

of the pore mouth for the oxygenated molecules, which could convert into aromatic compounds. The fresh catalyst had a surface area of 289.1 $m^2/g$, which reduced to 89.3 $m^2/g$ when 5 wt % of potassium was impregnated in the catalyst. However, the specific surface area of the catalyst with 5 wt % calcium impregnated was significantly higher at 132.4 $m^2/g$. The difference between the surface area of the catalysts deactivated by the same loadings of various metals could be due to their individual rate of diffusion into the pores of the zeolite structure. Stefanidis et al. [33] who studied hydrothermal deactivation and metal contamination of HZSM-5 catalysts reported a similar decrease in the surface area when they doped increasing amounts of inorganics (up to 9% of K, Ca, Mg and Na collectively). Similarly, Paasikallio et al. [35] reported a decrease in catalyst surface area from 212 to 118 $m^2/g$ after a 96 h CFP experiment with HZSM-5 catalyst. However, a major part of that loss in surface area was during the heat up phase and less pronounced during the course of the experiment. In catalytic pyrolysis experiments, these metals will be ejected from biomass particles and deposited on the catalyst surface thereby reducing the specific surface area and the acidity of the catalysts.

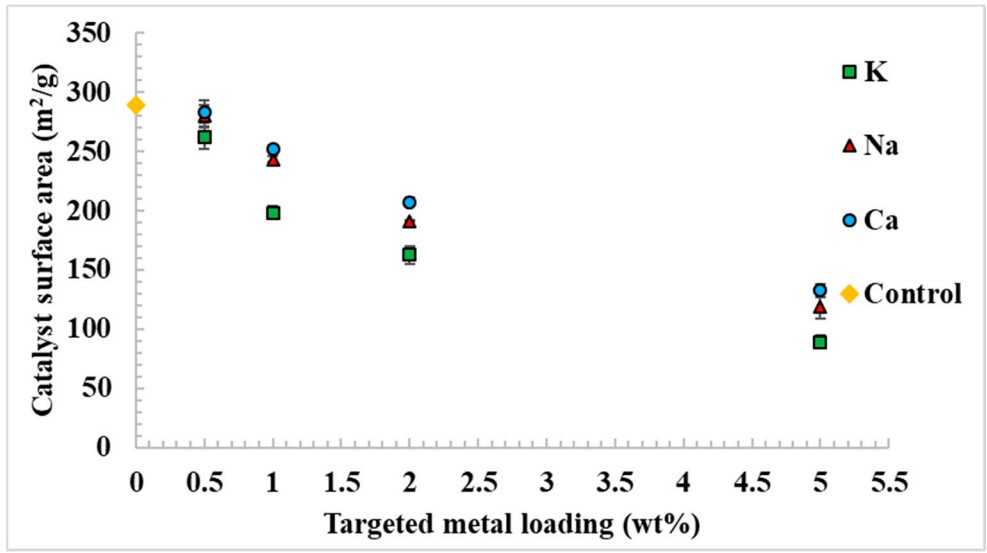

**Figure 1.** BET surface area of $H^+$ZSM-5 catalyst impregnated with biomass inorganics.

As mentioned earlier, inorganics added to the catalyst could cause physical and chemical poisoning. In physical poisoning, the active sites with proton functionality in the catalyst have less accessibility, whereas in chemical poisoning, those sites are rendered inactive. Chemisorption studies using ammonia ($NH_3$) as a probe molecule were performed to distinguish between physical and chemical deactivation. The $NH_3$-TPD profiles from different catalysts are shown in Figure 2, where the TPD curve for the fresh catalyst can be seen to exhibit two distinct desorption peaks at different temperatures: a lower temperature peak between 200 and 300 °C, and a higher temperature peak between 400 and 500 °C. The two peaks could be attributed to the presence of the weaker acid sites and the stronger acid sites in the $H^+$ZSM-5 catalyst, respectively. From the $NH_3$-desorption profiles, it can be clearly seen that the amount of strong acid sites is severely reduced by the presence of potassium and sodium, whereas calcium only reduces the amount of strong acid sites marginally, indicating that it does not cause significant chemical poisoning. In the case of potassium and sodium, a significant decrease in the peak corresponding to the weak acid sites could also be observed. Zheng et al. [37] observed the deactivation of the $V_2O_5$-$WO_3$-$TiO_2$ catalyst in a biomass combustion study and proposed that potassium could poison the Bronsted acid sites by proton-exchange and render them inactive for $NH_3$ adsorption, which is consistent with the TPD profile observed herein. Li et al. [38] studying the application of $H^+$ZSM-5 catalyst for propane dehydrogenation also observed a similar

decrease in the strong acid sites as a result of alkali metal impregnation on the catalyst. Paasikallio et al. [35] and Stefanidis et al. [33] also observed a negative correlation between increase in the accumulation of inorganics and the acidity of the catalyst.

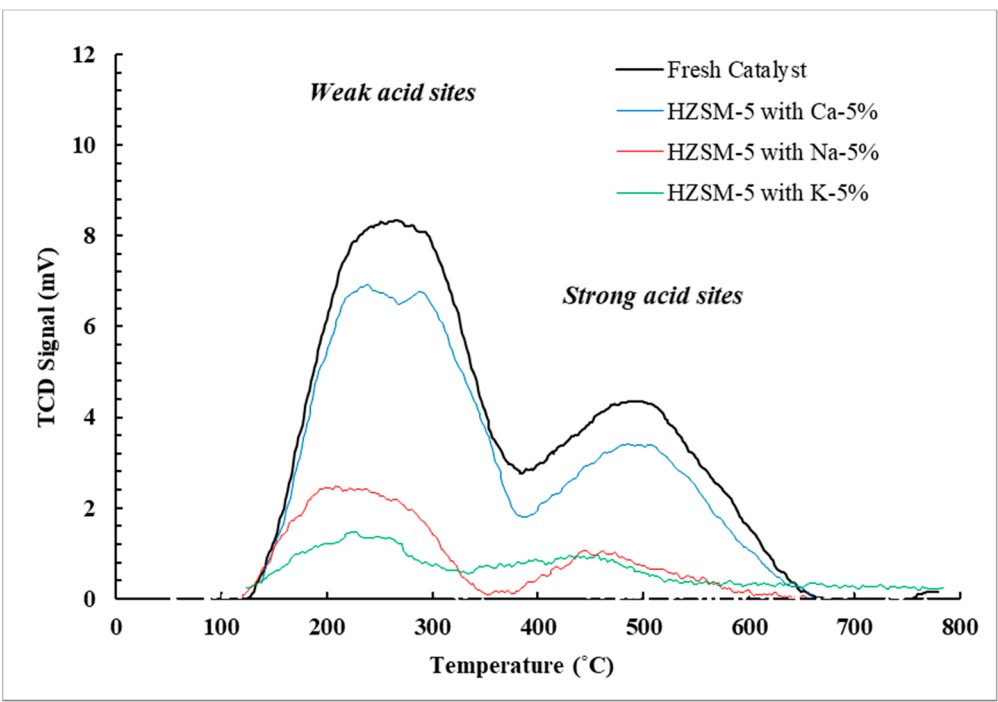

**Figure 2.** $NH_3$-temperature programmed desorption profiles from the $H^+$ZSM-5 catalysts deactivated with inorganic species.

### 2.3. Effect of Biomass Inorganics on In-Situ CFP

The fresh catalyst (control) and the catalysts deactivated by various levels of inorganic species were used as in-situ catalysts mixed with biomass in CFP experiments in a py/GC–MS setup. Major compounds identified from all the experiments were quantified using calibration standards and classified into five groups, listed in Table S1 (Supplementary Materials), along with the yields of individual compounds quantified in Tables S2–S4 (Supplementary Materials). For the sake of the discussion here, phenols, guaiacols, furans and ketones were further classified under the group of oxygenated compounds to compare with the aromatic hydrocarbons group.

The total carbon yield of all the major compounds observed and quantified from the CFP experiments is presented in Figure 3. It is clear that the presence of all the inorganic species resulted in a loss in carbon yield, which reduced significantly from 22.5% in the control experiments using the fresh catalyst to about 10.7% and 10.8% when the catalyst was deactivated by 5 wt % of potassium and sodium. The rate of decrease also appeared to be dramatic, reducing significantly with the presence of the 0.5 wt % and 1.0 wt % of potassium and sodium. Meanwhile, the presence of calcium at 0.5 wt % appeared to benefit the carbon yield initially showing a small increase when compared to the fresh catalyst. However, increase in the concentration of calcium beyond that caused a linear decrease in the total carbon yield. Progressive worsening of the carbon yield of condensable compounds could be related to the shift in product distribution due to the presence of the inorganics on the catalyst, which were known to promote the formation of solid residues (char and coke) and non-condensable gases by influencing the decomposition of cellulose, hemicellulose and lignin during pyrolysis [5,22,39]. However, it was not possible to confirm this known effect due to limitations of using a py-GC–MS. setup, which made it difficult to extract and measure the yield and carbon content of the solid residues consistently.

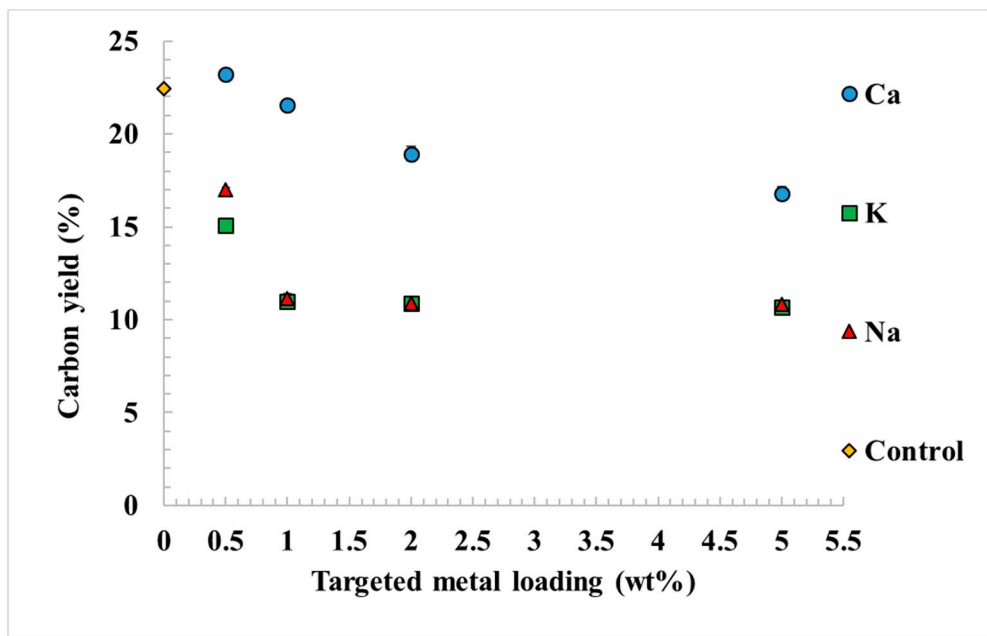

**Figure 3.** Total carbon yield from in-situ catalytic fast pyrolysis (CFP) experiments with H$^+$ZSM-5 catalysts deactivated by inorganics.

The role of using H$^+$ZSM-5 catalyst in upgrading pyrolysis vapor is to decrease the oxygen content of the resulting oil by producing hydrocarbon products from oxygenated compounds from cellulose, hemicellulose and lignin such as phenols, guaiacols, furans, ketones, acids and sugars such as levoglucosan. The Bronsted acid (H$^+$) sites on the catalyst, where H$^+$ binds to the negatively charged AlO$_4$$^-$ unit, acts as the active sites where the cracking reactions and decarboxylation, decarbonylation and dehydration reactions occur, resulting in the formation of the aforementioned hydrocarbons. In this study, the fresh catalyst (control) produced a carbon yield of 22.4% aromatic hydrocarbons as shown in Figure 4. Deactivation by calcium accumulation on the catalyst produced a linear decrease in the yield of aromatic hydrocarbons, reducing to 9.8% at the maximum loading of calcium added to the catalyst. Meanwhile, contamination by sodium and potassium appears to rapidly deactivate the catalyst, with 1 wt % of these alkali metals reducing the yield of aromatic hydrocarbons by more than 75% (from 22.4% to 4.9% and 5.5%, respectively). At the maximum loading of sodium and potassium, the production of aromatic hydrocarbons is completely suppressed and the condensable compounds identified in the GC–MS appeared to resemble the product composition from non-catalytic pyrolysis of biomass.

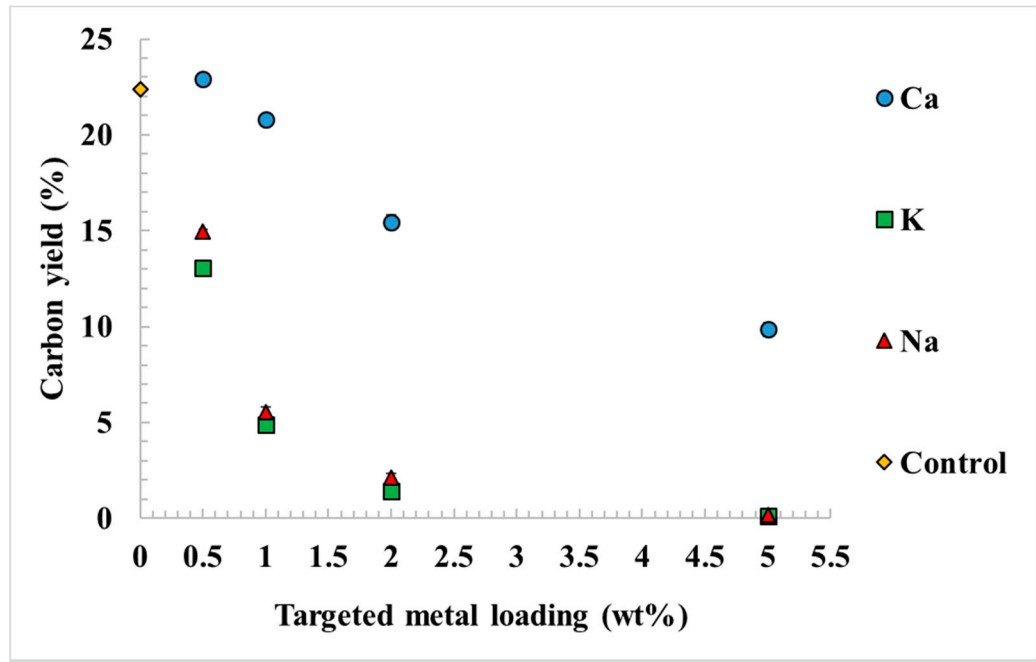

**Figure 4.** Aromatic hydrocarbons yield from in-situ CFP experiments with H$^+$ZSM-5 catalysts deactivated by inorganics.

The exponential rate at which this deactivation occurs is further confirmed by the evolution of oxygenated compounds with increasing levels of deactivation by inorganics. The yield of oxygenated compounds is expected to be inversely proportional to the activity of the catalyst, correlated well by the data from Figure 5. Several studies on in-situ CFP have also reported decreasing deoxygenation efficiency of the H$^+$ZSM-5 catalyst with increasing exposure to biomass ash [32–34,39]. In this study, CFP experiments with the fresh ZSM-5 catalyst resulted in complete deoxygenation of the pyrolysis vapor, resulting in all the products composed of monocyclic and polyaromatic hydrocarbons. Increasing concentrations of inorganics can be clearly seen to be shifting the product composition towards the formation of oxygenated compounds, with a linear increase observed again with calcium and an exponential increase in the case of potassium and sodium. At a concentration of 5 wt %, the catalysts deactivated by potassium and sodium yielded 10.6% of oxygenated compounds, with negligible amounts (<0.5%) of aromatic hydrocarbons. It has to be noted that the loss in the carbon yield of aromatic hydrocarbons is not directly compensated by the evolution of these oxygenated compounds and the loss in carbon yield could be a result of a change in product distribution towards the formation of coke and non-condensable gases. These results correlated well with the observations made from surface area and acidity characterizations of the catalysts. The changes due to deactivation by calcium appeared to primarily cause physical poisoning by limiting access to the active sites on the catalyst. The exponential rate at which the product composition changes due to sodium and potassium clearly shows the combined influence of physical poisoning and the inorganics affecting the active sites by removing the proton functionality of the catalyst.

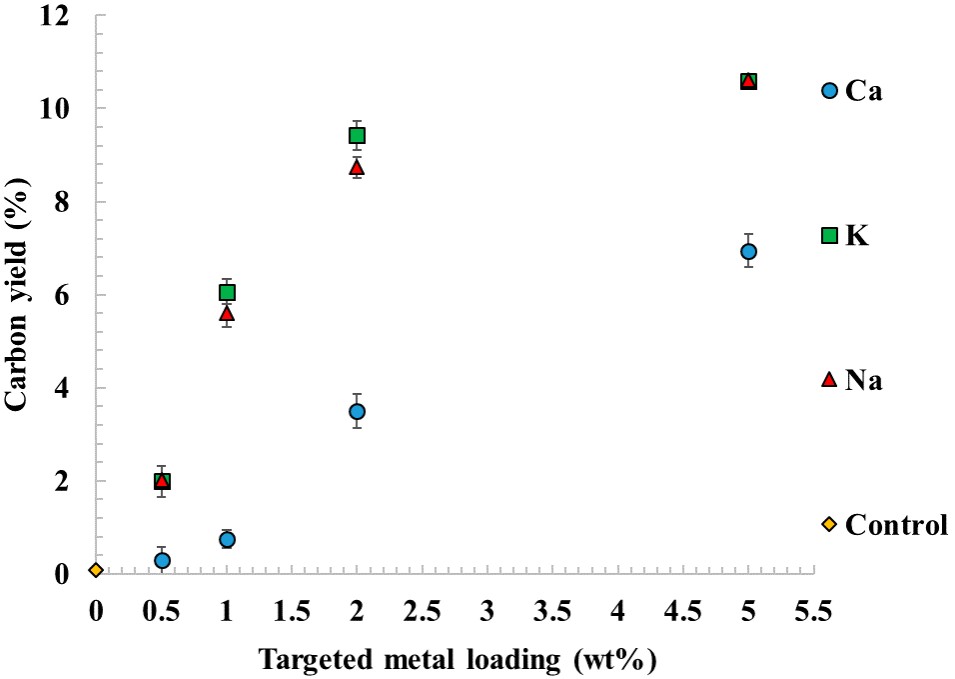

**Figure 5.** Oxygenated compounds yield from in-situ CFP experiments with H$^+$ZSM-5 catalysts deactivated by inorganics.

The major oxygenated compounds observed (guaiacols, phenols and furans) showed a clear increasing trend (Figure 6) as the concentration of inorganics on the catalyst increased. Furans formed from the decomposition of sugars in biomass have been known to be a precursor for the formation of aromatic hydrocarbons over zeolite catalysts. With increasing levels of deactivation by the inorganics, it is not surprising to observe that the yield of major compounds such as alkyl furans, furfural, 2(5H)-furanone and 2-furancarboxaldehyde, 5-methyl increased. The difference in the yield of furans between potassium-deactivated H$^+$ZSM-5 (4.33%) and calcium deactivated-H$^+$ZSM-5 (2.8%) could be another indicator that the calcium deactivated catalyst still retains some of the activity whereas the potassium-deactivated catalyst is completely inert. Similar changes in the yields of guaiacols and phenols were also observed, which increased with the increasing presence of the inorganics. However, the evolution of these compounds could also be a result of the catalytic activity of potassium and sodium on lignin, which was shown in a previous study from our group to enhance the depolymerization of lignin during pyrolysis and produce an increased yield of monomeric units such as phenols, alkylphenols and guaiacols [31]. The consequences of the deactivation of CFP catalysts due to inorganics observed in this study clearly demonstrated the need to consider the choice of biomass, pyrolysis reactor design and the robustness of the catalyst to prolonged exposure to inorganics found in biomass. Reactor designs that minimize the exposure of the catalyst to char and the inorganics, such as ex-situ upgrading, could potentially be the solution to avoid rapid catalyst deactivation.

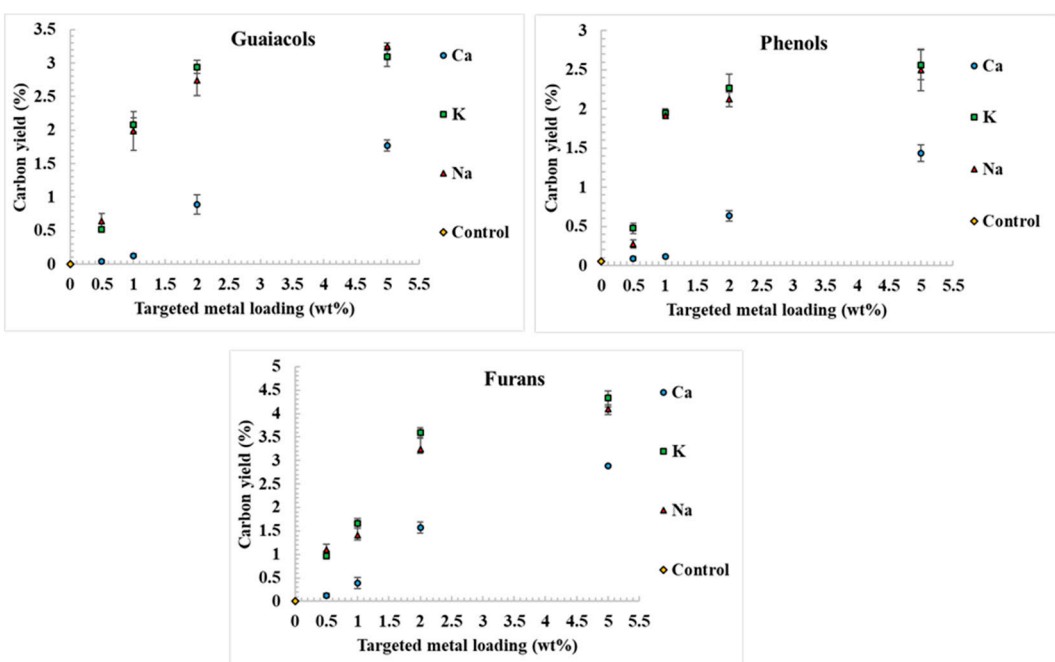

**Figure 6.** Yield of guaiacols, phenols and furans from in-situ CFP experiments with H+ZSM-5 catalysts deactivated by inorganics.

## 3. Materials and Methods

### 3.1. Materials

Southern pine wood chips obtained from a local wood chipping plant in Opelika, Alabama were used in this study. After an initial drying period of 72 h to reduce the moisture content of the wood chips, they were processed through a hammer mill (New Holland Grinder Model 358) fitted with a 1.58 mm (1/16 in.) screen. The sawdust obtained from the hammer mill was further sieved and the fraction that was smaller than 200 mesh (74 μm) was used for pyrolysis experiments in this study. The biomass was characterized for analyzing volatile matter, moisture and ash according to ASTM standards E871, E872 and E1755, respectively. The standard methods followed for determining the chemical constituents, inorganic content (ICP analysis), proximate and ultimate analyses of the biomass were described elsewhere [31].

The ammonium form of the ZSM-5 catalyst (CBV 3024, $SiO_2/Al_2O_3$ ratio of 30:1) was purchased from Zeolyst International (Conshohocken, PA, USA). The catalyst powder was sieved to remove coarse particles (>74 μm) and the fraction that was smaller than 200 mesh (74 μm) was used in this study. Four concentrations (0.5, 1, 2 and 5 wt %) of metals were impregnated in the ZSM-5 powder using the incipient wetness method. Briefly, appropriate quantities of the nitrates of Ca, K and Na were dissolved in 20 mL DI water according to the required concentration. The solution was stirred with the catalyst for 60 min, and subsequently dried at 120 °C to evaporate the solvent. The dried catalyst powder was then calcined in air at 550 °C for 5 h before use in pyrolysis experiments. The inorganic mineral content of the catalysts was measured after the impregnation by inductively coupled plasma (ICP) using a Thermo Scientific iCAP6300 ICP (Thermo Fisher Scientific, Waltham, MA, USA) spectrometer. A known amount of sample for ICP analysis was dissolved in concentrated $HNO_3$ and then diluted with 100 mL DI water. Calibration standards for Ca, Na and K were also prepared in the same background ($HNO_3$) and a four-point calibration curve was developed for each element. The BET (Brunauer–Emmett–Teller) surface area of the catalyst was measured using Quantachrome Autosorb-1 automated gas sorption system (Quantachrome Instruments, Boynton Beach, FL, USA). A known amount (0.20–0.30 g) of the catalyst sample was degassed at 300 °C under vacuum and then measured at

77.3 K (−195.85 °C) using nitrogen as the adsorbate. The strength and abundance of the acid sites on the catalyst were characterized using temperature-programmed desorption of ammonia. The characterization was performed in an Autosorb-iQ (Quantachrome Instruments, Boynton Beach, FL, USA), where a known amount of sample was degassed at 100 °C for 1 h of helium gas flow to remove the trapped water vapor, followed by flowing $NH_3$ gas (0.1 vol% in Ar) at 25 mL/min for 2 h at 40 °C. The saturated catalyst sample was then temperature programmed to 800 °C at a rate of 15 °C/min. To prepare the biomass/catalyst mixture for in-situ CFP experiments, 50 mg of the biomass and 150 mg of the catalyst were mixed using an ultrasonic bath (VWR Scientific, catalog no. 97043-960) to get a mixture having a biomass to catalyst ratio of 1:3. A microbalance with sensitivity of 0.001 mg (Metller Toledo, XP6) was used to measure the sample weight.

*3.2. Experimental Procedure—Pyrolysis GC–MS*

Pyrolysis experiments were performed in triplicates using a commercial pyrolyzer (Pyroprobe model 5200, CDS Analytical Inc., Oxford, PA, USA) connected to a gas chromatograph (Agilent Technologies, 7890A, Santa Clara, CA, USA). For each experiment, approximately 2 mg of the catalyst/biomass mixture was packed between quartz wool in the sample tube (1.9 mm I.D, 25 mm long) and placed in the pyrolysis chamber. The catalytic pyrolysis experiments were carried out at a filament temperature of 550 °C at a filament heating rate of 2000 °C/s and the filament temperature was held at 550 °C for 90 s. The interface temperature was maintained at 300 °C and was purged with helium gas flowing at a rate of 20 mL/min. The products from pyrolysis were absorbed by a trap maintained at 40 °C and these products were desorbed by heating the trap to 300 °C and transferred to the G.C. column through a transfer line and injector maintained at 300 °C and 250 °C, respectively. The condensable pyrolysis products were separated in the gas chromatograph using an Agilent DB 1701 capillary column (60 m × 0.250 mm and 0.250 μm film thickness). A split ratio of 1:100 was used for sample injection into the column, and the G.C. oven was programmed to start at 40 °C (hold time—3 min), after which it was ramped at 5 °C/min up to the final temperature of 270 °C (hold time—6 min). The column was connected to a mass spectrometer (Agilent Technologies, 5975MS, Santa Clara, CA, USA) for compound identification using the National Institute of Standard and Technology (NIST) mass spectral library and quantified by using calibration standards. Some of the major aromatic hydrocarbons identified were quantified using pure compounds purchased from Sigma-Aldrich (St. Louis, Mo, USA). Three different concentrations of the standards were prepared to obtain calibration factors for quantification.

Two factors of interest at various levels—(1) type of inorganic metal added to the catalyst (Ca, K and Na); and (2) amount of inorganic metal added to biomass (5000, 10000, 20000 and 50000 ppm or 0.5, 1.0, 2.0 and 5.0 wt %) were the focus of this study. The samples are labeled with their name followed by its metal loading in the catalyst. For example, Ca 0.5 refers to calcium 0.5 wt % loading in the catalyst. Results labeled as "control" are from experiments performed using catalyst without any added inorganics. Statistical analysis (ANOVA, Tukey's HSD) of the results was performed at 95% confidence interval using JMP software (SAS Institute, Cary, NC, USA). The product distribution from CFP experiments is reported in terms of carbon yield, which is the ratio of carbon in a specific product or group to the carbon contained in the feedstock. Selectivity of a particular aromatic hydrocarbon is defined as the ratio of moles of carbon in that product to the total moles of carbon in all aromatic hydrocarbons produced.

**4. Conclusions**

The accumulation of biomass inorganics and the resulting deactivation of the catalyst used in CFP is a critical issue that affects the commercial viability of the process. The effect of calcium, potassium and sodium on the deactivation of HZSM-5 was studied, and the inorganic species were found to affect its functionality by physical and chemical poisoning of the catalyst. All the inorganic species reduced specific surface area and accessibility to

the active sites, but sodium and potassium also chemically deactivated the catalyst by ion-exchange with the acid site ($H^+$) and reducing the strength of acid sites. The influence of deactivation by these metals on CFP was investigated in a py/GC–MS setup, which clearly indicated the strong negative influence on the performance of the catalyst. Accumulation of sodium and potassium even in very low concentrations (0.5 wt %) was found to be sufficient to cause an exponential loss in catalyst activity, whereas higher concentrations rendered the catalyst inert, losing its ability to deoxygenate pyrolysis vapor and produce aromatic hydrocarbons.

**Supplementary Materials:** A list of compounds identified in fast catalytic pyrolysis and product distribution in the presence of AAEMs are presented in the supplementary materials. This material is available free of charge via the Internet at https://www.mdpi.com/2073-4344/11/1/124/s1.

**Author Contributions:** Conceptualization: R.M. and S.A.; methodology: R.M., S.A. and R.S.; formal analysis: R.M. and R.S.; writing: original draft preparation: R.M.; writing: review and editing: S.A., R.S. and O.F.; project administration and funding acquisition: S.A. All authors have read and agreed to the published version of the manuscript.

**Funding:** This research was funded by National Institute of Food and Agriculture (USDA-NIFA-2015-67021-22842) and National Science Foundation (NSF-CBET-1333372). This work is part of the first author's requirements for the degree of Ph.D.

**Institutional Review Board Statement:** Not applicable.

**Informed Consent Statement:** Not applicable.

**Data Availability Statement:** Not applicable.

**Acknowledgments:** The authors would like to acknowledge the U.S. Department of Agriculture and the U.S. National Science Foundation.

**Conflicts of Interest:** The authors declare no conflict of interest.

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
