# Peer review of "Influence of Biomass Inorganics on the Functionality of H+ZSM-5 Catalyst during In-Situ Catalytic Fast Pyrolysis"

_catalysts, doi:10.3390/catal11010124_

Round 1

Reviewer 1 Report

This manuscript investigated the contamination of H+ZSM-5 catalyst by individual biomass inorganics including calcium, potassium, and sodium during in-situ CFP. During the CFP experiments, the authors used a micro reactor coupled with a GC/MS to get the product distribution from CFP, which helped them understand the influences of those inorganics. In the study, the reduction of carbon yield, aromatic hydrocarbons yield and the increase of oxygenated compounds yield are very clear. The conclusion is convincing and useful, but there are still my suggestions:

  1. Some supplementary experiments are recommended. For example, in 3.2, the authors said:” The difference between the surface area of the catalysts deactivated by the same loadings of various metals could be due to their individual rate of diffusion into the pores of the zeolite structure.” It would be better if there is some evidence to prove it.
  2. In Figure 3, the overlapped points are indiscernible.
  3. The conclusion can be improved, it is just a plain explanation of the work, not a highlighted summary of the conclusion. 

Author Response

This manuscript investigated the contamination of H+ZSM-5 catalyst by individual biomass inorganics including calcium, potassium, and sodium during in-situ CFP. During the CFP experiments, the authors used a micro reactor coupled with a GC/MS to get the product distribution from CFP, which helped them understand the influences of those inorganics. In the study, the reduction of carbon yield, aromatic hydrocarbons yield and the increase of oxygenated compounds yield are very clear. The conclusion is convincing and useful, but there are still my suggestions:

Thank you for the positive comments.

  1. Some supplementary experiments are recommended. For example, in 3.2, the authors said:” The difference between the surface area of the catalysts deactivated by the same loadings of various metals could be due to their individual rate of diffusion into the pores of the zeolite structure.” It would be better if there is some evidence to prove it.

Thank you for your comment. We have changed the word “deactivation” to “simply specific surface area reduction”. We do not know if the catalyst is in fact deactivated at that point. We simply loaded metals and measured the specific surface area. It is clear that there was a reduction in the surface area which will most likely translate into reduced catalyst activity. Thinking now, we should have also measured the pore volume and pore diameter of the catalysts. However, current situation does not allow us to run additional experiments to address this comment. In my mind, there is no doubt that the reduction in the specific surface area is due to pore blocking. There is nothing other than that.

  1. In Figure 3, the overlapped points are indiscernible.

This is true. I am not sure what to do with this because the data are overlapped.

  1. The conclusion can be improved, it is just a plain explanation of the work, not a highlighted summary of the conclusion. 

We appreciate the comments. We have highlighted the major findings in the conclusion. It is just that we have not provided the numbers since those were reported in the abstract. We did not want to repeat the same thing what was listed in the abstract.

Reviewer 2 Report

The paper reports on a very important problem of the influence of biomass composition onto the risk of catalyst deactivation. The paper is well organized, the introduction is very comprehensive and the presentation of the methods and results is also clear. In my opinion the paper can be accepted for the publication in Catalysts as it for sure is of the scientific importance and interest of the readers of this journal. However, prior to publication, I advice to make some minor corrections, as listed below.

1) Figure 1 - it would be better tu use yellow diamond (as in Fig. 3-6) for control case.

2) All figures would be more readable if you add ticks both on OX and OY axis.

3) Page 4, first paragraph of section 2.2: is the heating rate really so fast? (2000 deg. C/s)

4) Page 4, beginning of section 3.1: should be "higher heating value" not "heating value"

5) It would be good to add some further comments/conclusions on the effect of Ca which is quite different than the effect of K and Na. It might be related with the fact that Ca (and also Mg) generally increases the temperature of ash melting, thus in low content it can, in my opinion (look at so-called fuel or alkali indices), somehow, prevent the negative effect of the inherent ash (there is both K and Na in your biomass).  

Author Response

The paper reports on a very important problem of the influence of biomass composition onto the risk of catalyst deactivation. The paper is well organized, the introduction is very comprehensive and the presentation of the methods and results is also clear. In my opinion the paper can be accepted for the publication in Catalysts as it for sure is of the scientific importance and interest of the readers of this journal. However, prior to publication, I advice to make some minor corrections, as listed below.

Thank you for your positive comments. Also, we will address the minor comments raised by the reviewer.

1) Figure 1 - it would be better tu use yellow diamond (as in Fig. 3-6) for control case.

Changes have been made.

2) All figures would be more readable if you add ticks both on OX and OY axis.

Changes have been made.

3) Page 4, first paragraph of section 2.2: is the heating rate really so fast? (2000 deg. C/s)

Yes. This is a filament heating rate that uses to heat biomass, and I added “filament” to make it clear.

4) Page 4, beginning of section 3.1: should be "higher heating value" not "heating value"

Changes have been made.

5) It would be good to add some further comments/conclusions on the effect of Ca which is quite different than the effect of K and Na. It might be related with the fact that Ca (and also Mg) generally increases the temperature of ash melting, thus in low content it can, in my opinion (look at so-called fuel or alkali indices), somehow, prevent the negative effect of the inherent ash (there is both K and Na in your biomass).  

Yes, this could be a possibility but we saw the reduction in specific surface area and acidity when we doped our catalyst with these inorganic metals. We believe the effect of K and Na could be even more under actual situation.